# LDOC1 as Negative Prognostic Marker for Vulvar Cancer Patients

**DOI:** 10.3390/ijms21239287

**Published:** 2020-12-05

**Authors:** Giulia Wanka, Elisa Schmoeckel, Doris Mayr, Sophie Fuerst, Christina Kuhn, Sven Mahner, Julia Knabl, Maria Margarete Karsten, Christian Dannecker, Helene H. Heidegger, Aurelia Vattai, Udo Jeschke, Julia Jueckstock

**Affiliations:** 1Department of Obstetrics and Gynecology, University Hospital, LMU Munich, Marchioninistraße 15, 81377 Munich, Germany; Giulia.Wanka@campus.lmu.de (G.W.); Sophie.Fuerst@med.uni-muenchen.de (S.F.); Christina.Kuhn@med.uni-muenchen.de (C.K.); Sven.Mahner@med.uni-muenchen.de (S.M.); Julia.Knabl@med.uni-muenchen.de (J.K.); Helene.Heidegger@med.uni-muenchen.de (H.H.H.); Aurelia.Vattai@med.uni-muenchen.de (A.V.); julia.knabl@gmx.de (J.J.); 2Department of Pathology, LMU Munich, Thalkirchner Str. 142, 80337 Munich, Germany; Elisa.Schmoeckel@med.uni-muenchen.de (E.S.); Doris.Mayr@med.uni-muenchen.de (D.M.); 3Department of Obstetrics, Klinik Hallerwiese, Sankt-Johannis-Mühlgasse 19, 90419 Nürnberg, Germany; 4Department of Gynecology and Gynecologic Oncology, Charité University, Charitéplatz 1, 10117 Berlin, Germany; Maria-Margarete.Karsten@charite.de; 5Department of Obstetrics and Gynecology, University Hospital Augsburg, Stenglin Street 2, 86156 Augsburg, Germany; Christian.Dannecker@uk-augsburg.de

**Keywords:** vulvar carcinoma, LDOC1, cancer survival, prognosis, NF-κB, C-DIM 12

## Abstract

So far, studies about targeted therapies and predictive biomarkers for vulva carcinomas are rare. The leucine zipper downregulated in cancer 1 gene (LDOC1) has been identified in various carcinomas as a tumor-relevant protein influencing patients’ survival and prognosis. Due to the lack of information about LDOC1 and its exact functionality, this study focuses on the expression of LDOC1 in vulvar carcinoma cells and its surrounding immune cells as well as its correlation to clinicopathological characteristics and prognosis. Additionally, a possible regulation of LDOC1 in vulvar cancer cell lines via the NF-κB signaling pathway was analyzed. Vulvar carcinoma sections of 157 patients were immunohistochemically stained and examined regarding LDOC1 expression by using the immunoreactive score (IRS). To characterize LDOC1-positively stained immune cell subpopulations, immunofluorescence double staining was performed. The effect of the NF-κB inhibitor C-DIM 12 (3,3′-[(4-chlorophenyl)methylene]bis[1 H-indole]) on vulvar cancer cell lines A431 and SW 954 was measured according to MTT and BrdU assays. Baseline expression levels of LDOC1 in the vulvar cancer cell lines A431 and SW 954 was analyzed by real-time PCR. LDOC1 was expressed by about 90% of the cancer cells in the cytoplasm and about half of the cells in the nucleus. Cytoplasmatic expression of LDOC1 was associated with decreased ten-year overall survival of the patient, whereas nuclear staining showed a negative association with disease-free survival. Infiltrating immune cells were mainly macrophages followed by regulatory T cells. Incubation with C-DIM 12 decreased the cell viability and proliferation of vulvar cancer cell line A431, but not of cell line SW 954. LDOC1 expression on mRNA level was twice as high in the cell line A431 compared to the cell line SW 954. Overexpression of LDOC1 was associated with unfavorable overall and disease-free survival. Tumor growth could be inhibited by C-DIM 12 in vitro if the expressed LDOC1 level was high enough.

## 1. Introduction

Vulvar cancer is a rare female malignancy making up about 5% of all gynecologic cancers and affecting mostly elderly (>70 years) women [1]. However, over the past few decades, the incidence of vulvar cancer and vulvar intraepithelial neoplasia (VIN) has reportedly increased, particularly among younger (<50 years) women [2,3]. Squamous cell carcinomas account for >90% of the malignant tumors of the vulva [4]. There are two distinct aetiopathogenic pathways leading to vulvar squamous cell carcinoma (VSCC): one type of VSCC, accounting for approximately 40% of carcinomas, is human papillomavirus (HPV)-driven and affects mostly young women [5]. The associated premalignancy is called vulvar intraepithelial neoplasia of usual type (uVIN). The other type, accounting for the remaining 60%, is more prevalent in older patients and is associated with lichen sclerosus and/or differentiated vulvar intraepithelial neoplasia (dVIN) as premalignant lesions [6]. Several morphological variants have been described, including basaloid, warty, and keratinizing carcinoma, the latter being the most common. Keratinizing variants are considered to be HPV-negative, whereas basaloid and warty squamous cell carcinomas are more likely to be related to human papillomavirus (HPV) [6]. HPV-related malignancies are associated with a persistent HPV infection [7]. The host immune response is of crucial importance in determining the clearance or persistence of HPV infections and HPV-related VIN. In immuno-compromised patients, the immune system cannot clear a transient HPV infection. Nicotine consumption additionally suppresses the immune system [8]. In fact, 60% of the patients with vulvar carcinoma smoke or have smoked [9]. Other than HPV infection, immunosuppression and smoking, chronic inflammatory dermatosis as lichen sclerosus and metasynchronous genital dysplasia are important risk factors for developing vulvar cancer [10]. Conventional therapy of vulvar cancer consists of radical excision and/or (chemo-)radiation. The major obstacles in the successful treatment of vulvar cancer are lymph node invasion and high recurrence rates, leading to decreased survival [11,12]. Therefore, target-orientated therapy strategies addressing specific tumorigenic pathways and predictive biomarkers are needed to improve the clinical outcome of patients with advanced disease, highly affected lymph nodes, or cancer recurrence.

Checkpoint inhibitors represent a possible therapeutic approach, being effective in a variety of cancers, leading to a strong immune response against tumor cells by blocking programmed death-ligand 1 (PD-L1) [13]. PD-L1 has become a valid biomarker that is routinely analyzed in several types of cancer as a predictive indicator for therapy with checkpoint inhibitors [14]. Interestingly, several studies could demonstrate that PD-L1 expression is related to HPV status, suggesting that PD-L1 expression is increased in HPV-associated carcinomas such as vulvar cancer [13,15]. In fact, a recent study from Czogalla et al. showed that PD-L1 expression is frequent in VSCC and high PD-L1 expression was associated with an unfavorable outcome [16].

In addition to biomarkers like PD-L1, tumor suppressors can have an influence on cancer prognosis and survival via their expression patterns. In this study, we focused on leucine zipper downregulated in cancer 1 (LDOC1), a tumor suppressor candidate gene that was discovered for the first time to be downregulated in pancreatic and gastric cancer cells [17]. LDOC1 is a small transcription factor of only 146 amino acids (17 kDa) and contains an eponymous DNA-binding leucine zipper motif and a putative protein–protein interacting, proline-rich region [17]. LDOC1 is predominantly expressed in the nucleus of cells and can be found ubiquitously in the human body. It plays an important role in modulating cell proliferation via the NF-κB signaling pathway [18]. Downregulation of LDOC1 was shown in various tissue samples like colorectal cancer [19], papillary thyroid cancer [20] and cervical [21] and ovarian cancer [22]. Buchholtz et al. revealed that LDOC1 expression is epigenetically regulated in cervical and ovarian cancer by promoter methylation [21,22]. However, studies examining other tumor identities such as chronic lymphatic leukemia (CLL) [23] or head and neck squamous cell carcinoma [24] come to opposite conclusions. Those studies have reported that LDOC1 overexpression increases cell proliferation and is correlated with a poor prognosis [23,25]. Since no information about the expression and involvement of LDOC1 in vulvar cancer exists to date, we analyzed the expression of LDOC1 in tissue samples and vulvar cancer cell lines.

## 2. Results

### 2.1. Study Group and Clinical Data

The age of the examined patients ranges from 20 to 96, with a mean age of 67 years. About 90% of the patients showed a keratinizing VSCC, and the remaining 10% a warty/basaloid VSCC. Further clinicopathological parameters are listed in Table 1. Follow-up examinations were carried out with a mean follow-up time of 71.4 months and a range from 0 to 276.7 months. Overall, 121 out of the 177 patients (68%) died during the follow-up period.

### 2.2. LDOC1 Expression in Vulvar Cancer Tissue

The LDOC1 staining process in the cytoplasm resulted in a weak color reaction in 27.6% of the vulvar carcinoma tissue sections (IRS 1–2) (Figure 1a), 52.7% showed a moderate color reaction (IRS 3, 4, 6), and 9.5% of the specimens showed a strong color reaction (IRS 8, 12). In 10.2%, LDOC1 was not expressed in the cytoplasm (IRS 0). Regarding the nuclear expression of LDOC1 (Figure 1b), 11% of the vulvar carcinoma tissue sections showed a weak color reaction (IRS 1–2), 33.1% showed a moderate color reaction (IRS 3, 4, 6), and 3.1% of the specimens showed a strong color reaction (IRS 8). In 52.8% of patients, LDOC1 was not expressed in the cell nucleus (IRS 0). The average IRS for the LDOC1 expression in the cytoplasm was 3.15 (total range 0–12), whereas the expression in the nucleus resulted in a mean value of 1.65 (total range 0–8). In order to investigate if and how these two parameters are related, a correlation analysis using the Spearman correlation function was performed. This showed a significant (*p* = 0.000–0.028, ρ = 0.412) positive correlation, indicating that the higher the IRS is in the cytoplasm, the higher it is in the nucleus. The positive correlation curve of the LDOC1 IRS in the cytoplasm and the cell nucleus is shown in Figure 2.

There were no significant correlations between LDOC1 expression and clinicopathological characteristics of the analyzed vulvar carcinoma samples listed in Table 1, including the HPV status which was assessed using its surrogate marker p16.

In normal vulvar tissue, no expression of LDOC1 was shown, whereas moderate cytoplasmatic expression of LDOC1 could be detected in vulvar intraepithelial neoplasia (Appendix A).

### 2.3. LDOC1 Expression and Patients’ Survival

One-dimensional survival analyses using a log-rank test showed that positive expression of LDOC1 in the cytoplasm (IRS > 2) was significantly correlated with a reduced ten-year overall survival (*p* = 0.049; Figure 3a). For the cases where the cytoplasmatic IRS was larger than two, the red line shows a significantly lower cumulative survival than the survival for the cases with a cytoplasmatic IRS less than two, illustrated by the blue line. Additionally, samples with more than 55% of the cells being cytoplasmatic positively stained were significantly related to a worse ten-year overall survival (*p* = 0.046; Figure 3b), illustrated in red. Samples with less than 55% positively stained cells, illustrated in blue, displayed a significantly higher cumulative survival than the opposing scenario. Prognostic information regarding overall survival was only evident in cytoplasmatic positively stained cells, while nuclear staining did not provide significant results regarding the overall survival (Appendix A). Looking at the ten-year disease-free survival, however, we found that nuclear staining of LDOC1 (IRS > 1) showed a significant negative effect (*p* = 0.031; Figure 3c). The red curve in Figure 3c represents the ten-year disease-free survival for samples with a nuclear IRS larger than one, which lies below the blue line representing the contrasting case of samples with a nuclear IRS of less than one.

### 2.4. LDOC1 Expression on Infiltrating Immune Cells

In 96% of the specimens, LDOC1-positive immune cells could be detected. There was no significant influence of LDOC1-positive immune cells on clinicopathological parameters nor patients’ survival. In order to characterize the immune cells subpopulations, immunofluorescence double staining for LDOC1 and three markers for leucocyte subpopulations, CD56, CD68 and FOXP3, was carried out. In Figure 4, representative photomicrographs of the immunofluorescence double staining are presented. The left column shows LDOC1-positive cells (red), the column in the center shows the marked immune cell (green), and the right column merges the two expression patterns to identify a possible co-expression of LDOC1 (red) and stained immune cells (green). Double staining of LDOC1 and CD56 as a marker for natural killer cells showed no result (Figure 4a). No CD56-positive immune cells could be detected in the tumor bed infiltrated by immune cells as displayed in the center column of Figure 4a, whereas double staining of LDOC1 and CD68 (Figure 4b) as a marker for macrophages as well as FOXP3 (Figure 4c) as a marker for regulatory T cells showed positive results, as illustrated in the Figure 4b,c by green stained cells. The immune cell subpopulations were quantified by counting CD56-, CD68-, and FOXP3-positive cells per field of view (20x magnification). Most infiltrating cells were CD68-positive macrophages, followed by FOXP3-positive regulatory T cells.

### 2.5. Regulation of LDOC1 by Stimulation with C-DIM 12

To evaluate the potential regulation of the expression of LDOC1 on vulvar cancer cell lines, MTT- and BrdU-assays were carried out after incubation with the NF-κB inhibitor C-DIM 12 (3,3′-[(4-chlorophenyl)methylene]bis [1H-indole]). The MTT-assay showed a decreased cell viability of A431 cells starting at a concentration of 1 μM of C-DIM 12 compared to the unstimulated control cells (Figure 5a). The decrease in the optical density of the respective concentrations compared to the control group was significant (*p* = 0.005–0.013). However, no influence on cell viability could be detected for the SW 954 cell line (Figure 5b). Since no effect on cell viability could be detected in the SW 954 cell line after stimulation with C-DIM12, the BrdU-assay, which investigates cell proliferation, was not performed with cell line SW 954. For cell line A431, the BrdU-assay showed that the addition of C-DIM 12 and incubation for 72 h had an adverse effect on cell proliferation (Figure 5c). The statistical evaluation showed a significant decrease in optical density with respect to the control group (*p* = 0.002–0.003). To test why the expression of LDOC1 could be regulated in vulvar cancer cell line A431 and not in vulvar cancer cell line SW 954, quantitative real-time PCR was performed to analyze the relative expression of LDOC1 in the two vulvar cancer cell lines. In the A431 cell line, *LDOC1* expression was twice as high compared to the cell line SW 954 (Figure 5d). The difference between the two cell lines was found to be statistically significant with *p* = 0.026.

## 3. Discussion

In the present study, we investigated the expression of LDOC1 in vulvar cancer tissue and infiltrating immune cells and a further regulation of LDOC1 in vitro. LDOC1 was identified as a negative prognostic marker in vulvar cancer by immunohistochemical staining, showing a negative association with patients’ disease-free survival as well as overall survival. LDOC1 was expressed in about 90% of the tissue samples in the cytoplasm and within 50% in the nucleus. Interestingly, LDOC1 was significantly more highly expressed in the cytoplasm with a mean IRS of 3.15 compared to a mean IRS of 1.65 in the nucleus. The cytoplasmic expression of LDOC1 was associated with significantly decreased ten-year overall survival, whereas nuclear expression was associated with unfavorable ten-year disease-free survival.

Since LDOC1 is ubiquitously expressed in normal brain, thyroid, heart, kidney and pancreas tissue but is downregulated in many tumors, the hypothesis of LDOC1 as a tumor suppressor gene was formulated by Nagasaki et al. [17]. In addition to tumor identities such as pancreatic [26], esophageal [27] and colorectal cancer [19], a downregulation of LDOC1 has also been found in cervical and ovarian cancer. The silencing of the LDOC1 gene by promoter methylation has been identified as the responsible mechanism [21,22]. LDOC1 is considered to be a protein which is mainly expressed in the cell nucleus [17]. However, in the patient population of this study, less than half of the tissue samples expressed LDOC1 in the nucleus. Tumor suppressor proteins are known to lose their suppressing function upon exiting their original localization and translocation into the cytoplasm. Duzkale et al. revealed that upregulation of LDOC1 in unmutated chronic lymphatic leukemia (CLL) was associated with a poor prognosis [18]. The high mRNA levels of LDOC1 found in that study could not be explained either by changes in the copy number of the gene or by mutations in the coding region. The authors therefore hypothesized that LDOC1 complexes with LDOC1S, a splice variant of LDOC1, and, thus, loses its antiproliferative function by forming a dysfunctional dimer in the nucleus [23]. Previous to this study, similar results had already been found by Mizutani et al., which identified WAVE3, a protein belonging to the Wiskott–Aldrich syndrome protein family, as a negative regulator of LDOC1. Upon co-expression of WAVE3 and LDOC1, the pro-apoptotic function of LDOC1 is inhibited by translocation of LDOC1 from the nucleus to the cytoplasm [28]. Moreover, sex-specific environmental mechanisms can mediate the loss of LDOC1 function as a tumor suppressor. A recent study comparing saliva samples from patients with oral squamous cell carcinoma revealed a gender-specific difference in LDOC1 expression: while LDOC1 expression levels were low in males, LDOC1 expression levels were considerably higher in female patients [29].

In contrast to the LDOC1-positive area of the tumor, LDOC1-positive infiltrating immune cells did not have any influence on the patients’ outcome. Interestingly, we found that in almost all samples, the immune cells were expressing LDOC1. Infiltrating immune cells can either inhibit or promote tumor progression depending on their populations [30]. Consequently, we wanted to characterize the immune cells that expressed LDOC1. Immunofluorescence double staining identified CD68-positive macrophages as the main part of the LDOC1-positive immune cell infiltrate, followed by FOXP3-positive regulatory T cells. Concordantly, Abdulrahman et al. showed that VSCC and their precursors are infiltrated with variable numbers of macrophages and regulatory T cells, indicating that they express targetable tumor antigens [31]. Tumor-associated macrophages (TAM) are known to facilitate tumor progression by increasing cancer cell migration and invasiveness, stimulating angiogenesis and suppressing anti-tumor immunity [32]. Especially in solid tumors, like vulvar carcinoma, macrophages produce cytokines and growth factors interacting with important signaling pathways [33]. These findings suggest that macrophages may play an important role in tumor microenvironment of vulvar carcinoma, attributing to an immunosuppressive effect. Since macrophages can have a negative influence on the tumor, it seems plausible that they express LDOC1, as it was identified as a negative prognostic marker for vulvar cancer in this study. Since very little is known about LDOC1 and its expression in immune cells, our study serves the purpose of presenting a general assessment to this hypothesis. Whether the expression of LDOC1 in the tumor attracts mainly macrophages, or whether the tumor suppressor LDOC1 is expressed independently in the immune cells, must be investigated in following experiments. In addition, a further analysis of the microenvironment is important for immunotherapeutic strategies in vulvar carcinoma regarding a possible checkpoint inhibition therapy.

Furthermore, LDOC1 is thought to regulate the transcriptional response mediated by the nuclear factor kappa B (NF-κB). By inhibiting apoptosis and promoting proliferation, NF-κB interferes with the balance between proliferation and apoptosis in favor of the malignant growth of tumor cells [34]. In many tumor identities, the NF-κB signaling pathway is activated and associated with poor prognosis [35]. Activation of NF-κB by growth inhibitory cytokines could also be detected in the nucleus of vulvar cancer cells [36]. A study by Song et al. revealed that upon LDOC1 transfection of human intrahepatic biliary epithelial cells (HIBECs), the mRNA rate of NF-κB increased significantly while the apoptosis rate decreased [25]. The authors concluded that LDOC1-mediated translocation of NF-κB into the cell nucleus inhibits programmed cell death by activating gene expression of downstream inflammatory mediators [25]. Since the signaling pathway around NF-κB plays a crucial role in vulvar carcinomas [36] and is regulated by LDOC1, the effect of a possible inhibitor C-DIM 12 on the vulvar carcinoma cell lines A431 and SW 954 was investigated in the present study. Inamoto et al. primarily investigated the effect of 1,1-bis(3′-indolyl)-1-(p-chlorophenyl)methane (C-DIM), a Nurr1 activator, that via inhibiting the NF-κB signaling pathway reduced tumor growth of bladder cancer cells in vitro [37]. We found that increasing concentrations of C-DIM 12 starting at a concentration of 1 µM caused a significant decrease in the cell viability measured in the MTT assay after incubation of the cell line A431 for 72 h. The proliferation rate of the cell line A431 measured by BrdU assay also decreased after stimulation with C-DIM 12, but already starting at a concentration of 0.1 µM. We concluded that the stimulant C-DIM 12 increases the rate of apoptosis and reduces proliferation in vulvar cancer by inhibiting NF-κB. A further analysis is required to investigate the C-DIM 12 efficiency depending on different concentrations of C-DIM 12. However, no decrease in cell viability could be detected for the cell line SW 954 by stimulation with C-DIM 12. We then investigated the basal LDOC1 expression rate of both cell lines using real-time PCR. The PCR revealed that the relative expression rate of LDOC1 in the A431 cell line was twice as high as that of the SW 954 cell line. Thus, C-DIM 12 acts via indirect LDOC1 regulation on the A431 cell line, which expresses a larger amount of LDOC1. By inhibiting the NF-κB signaling pathway, C-DIM 12 is able to reduce tumor growth in vulvar cancer cells in vitro.

Since NF-κB is an important transcription factor, LDOC1 attacks a central mechanism of gene regulation. The clinical significance lies in the possibility to develop specific drugs that either minimize the expression of LDOC1 itself or intervene later in the downstream signaling pathway by inhibiting NF-κB.

## 4. Materials and Methods

### 4.1. Study Group and Clinical Data

The study group consisted of 177 patients with vulvar carcinoma, primarily diagnosed between 1990 and 2008, treated at the department of Gynecology and Obstetrics of the Ludwig-Maximilians-University in Munich, Germany. Tissue samples were surgically obtained and histologically processed as part of routine clinical care. Patients’ clinical data were collected, and follow-up examinations were carried out. Of the 177 tissue samples, 157 were available for immunohistochemical staining.

### 4.2. Ethical Approval

All patients’ data were fully anonymized, and the study was performed, according to the standards set in the Declaration of Helsinki 1975. The tumor tissue used was leftover material that had initially been collected for histopathological diagnostics. All diagnostic procedures have already been fully completed when samples were retrieved for the study. The current study was approved in writing by the Ethics Committee of the Ludwig-Maximilians-University, Munich, Germany (approval number 19–261). Authors were blinded for clinical information during experimental analysis.

### 4.3. Immunohistochemistry

Immunohistochemical stains were performed using formalin-fixed paraffin-embedded (FFPE) tissues. Sections were cut at 4 µm from each paraffin block and mounted on SuperFrost Plus microscope slides (Menzel Glaeser, Braunschweig, Germany). Tissue specimens were deparaffinized in xylol for 20 min, followed by washing in 100% ethanol. Endogenous peroxidase activity was inhibited with 3% hydrogen peroxide diluted in methanol for 20 min before rehydrating the slides in a descending alcohol series (100%, 70%, 50%). After rinsing with distilled water, the slides were heated with citric acid buffer in a pressure cooker to unmask the antigens’ epitopes. Subsequently, sections were rinsed two times with phosphate buffered saline (PBS). Blocking and antibody staining procedures were performed using Zytochem-Plus HRP Polymer-kit (Zytomed, Berlin, Germany). After incubating with the blocking solution for 5 min, samples were incubated with the polyclonal rabbit IgG anti-LDOC1 antibody (Abcam, Cambridge, UK), which was diluted at the ratio of 1:100, for 16 h (h) at 4 °C in a humid chamber. Post-block reagent was applied, and the samples were incubated for 20 min at room temperature in a humid chamber. HRP-Polymer was then applied and incubated for 30 min. The addition of 3,3′-diaminobenzidine (Dako, Hamburg, Germany) started the peroxidase substrate staining reaction, which led to the color precipitation that could be seen with a light microscope. After each incubation with the primary antibody, post-block and HRP-Polymer, the slides were rinsed two times with PBS. In the end, slides were counterstained with hemalum, dehydrated, and covered. To support validity of the LDOC1 staining, slides made from placenta tissue were used as positive control.

Specimens were evaluated under the light microscope (Leitz, Wetzlar, Germany) using the semi-quantitative immunoreactive Score (IRS) by Remmele and Stegner [38]. LDOC1 showed two different expression patterns in the evaluation; therefore, its cytoplasmic and nuclear staining were independently evaluated by IRS. Patients’ data were correlated with the IRS itself as well as the two individual constituent parameters of the score: optical staining intensity and percentage of positively stained cells.

### 4.4. Immunofluorescence

To characterize the immune cell subpopulations, immunofluorescence double staining for LDOC1 and CD68 (Sigma Aldrich, St. Louis, MO, USA), CD56 (Serotec, Oxford, UK) as well as FOXP3 (Abcam, Cambridge, UK), representing markers for leucocyte subpopulations, was performed. The slides were pretreated, comparing the immunohistochemistry procedure. The samples where then incubated for 15 min with the Ultra Vision Protein Block (Thermo Scientific, Lab Vision, Fremont, CA, USA) to prevent unspecific binding of the primary antibody. A mixed solution of the primary antibodies was added at the respective concentrations: LDOC1 and CD56 were diluted 1:100, CD68 1:800 and FOXP3 1:300. The slides were incubated for 16 h at 4 °C in a humid chamber. After washing the slides three times with PBS, they were covered for 30 min in the dark with the fluorophore-labeled secondary antibodies Goat-Anti-Rabbit IgG Cy3 (Dianova, Hamburg, Germany) and Goat-Anti-mouse-AlexaFlour488-IgG (Dianova, Hamburg, Germany). Finally, the slides were rinsed three times with PBS and covered with mounting medium (Vectashield H-1200; Vector Laboratories, Burlingame, CA, USA) containing DAPI for nuclear counterstaining.

Specimens were evaluated under the laser microscope (Axiophot fluorescent microscope; Zeiss, Oberkochen, Germany) using the corresponding software AxioVision (Rel. 4.8, Zeiss, Oberkochen, Germany).

### 4.5. Cells and Cell Culture

The vulvar cancer cell line A431 was purchased from ECACC (European Collection of Cell Cultures, Salisbury, UK), and the vulvar cancer cell line SW 954 was purchased from ATCC (American Type Culture Collection, Manassas, VA, USA). Cells were cultured in Dulbecco’s modified Eagle’s medium (DMEM) (Biochrom, Berlin, Germany) supplemented with 10% fetal bovine serum (PAA, Pasching, Austria) at 37 °C in a humidified atmosphere with 5% CO_2_.

### 4.6. Stimulation with C-DIM 12

The A431 and SW 954 cells were seeded for the following experiments with a density of 5000 cells/100 μL/well in a sterile 96-well-plate and incubated overnight. After 24 h, the stimulant C-DIM 12 (Tocris, Bio-Techne Corporation, Minneapolis, MN, USA) was added at the respective concentrations (0.2, 2, 20 µM) and the cells were then incubated for 72 h. Then, 100 μL of each of the prepared solutions were pipetted to the cells. The 1:1 dilution in the well gave a final concentration of 0.1, 1, and 10 μM. The concentrations used are based on previous studies with C-DIM 12 [39,40]. Unstimulated cells served as a negative control.

### 4.7. MTT-Assay

Cell viability after stimulation with C-DIM 12 for 72 h was evaluated using the MTT assay. Cell lines A431 and SW 954 were cultured in 200 µL growth medium in 96-well-plates. After incubation with C-DIM 12, 20 µL of MTT solution (5 mg/mL in PBS; Sigma-Aldrich, St. Louis, MO, USA) were added and cells were incubated for 1.5 h at 37 °C in a 5% CO_2_ humidified atmosphere. After forming blue crystals, the cell culture medium was removed, and crystals were solubilized by adding 200 µL dimethyl sulfoxide (DMSO) per well. Absorbance was measured at a primary wavelength of 595 nm by ELISA-Reader (BIO-TEK Instruments GmbH, Bad Friedrichshall, Germany) and Gen 5 Software (BIO-TEK Instruments GmbH, Bad Friedrichshall, Germany). All values were evaluated in triplets and the test was repeated three times to confirm its reliability.

### 4.8. BrdU-Assay

The colometric bromodeoxyuridine (BrdU) assay detecting proliferating cells was performed with the BrdU kit from Roche (Rotkreut, Switzerland), which consists of 6 substances (bottles 1–6). The cell line A431 was cultured in the presence of C-DIM 12 for 72 h in a 96-well-plate. Subsequently, BrdU was added and the cells were reincubated for 24 h. After removing the culture medium, Fix-Denat was added to denature the DNA. Afterwards, anti-BrdU-POD solution was added, which binds to the BrdU incorporated in newly synthesized DNA. After incubation with the antibody solution, the supernatant was removed, and the wells were washed three times with a washing solution. To detect the immune complexes, the substrate solution was added and incubated in the dark at room temperature. An increasing deep blue color became visible after about 20 min. The discoloration was stopped by adding sulfuric acid. Finally, the plate was measured at a wavelength of 450 nm by ELISA-Reader using Gen 5 Software. All values were evaluated in triplets and the test was repeated three times.

### 4.9. RNA Isolation and cDNA Synthesis

The principle of RNA isolation consists of three main steps: cell lysis, inactivation of RNases and isolation of RNA. The purification of RNA from A431 and SW 954 was performed using the RNeasy Mini Kit (QIAGEN, Hilden, Germany). Therefore, half a million cells per cell line were seeded in a 6-well-plate and incubated overnight. At the end of the incubation period, RLT buffer diluted with β-mercaptoethanol (Sigma-Aldrich, St. Louis, MO, USA) was added to the cells. The RLT buffer disrupts the cells, while ß-mercaptoethanol inactivates RNases. The cell lysates were then placed on a QIAshredder spin column and centrifuged for 2 min to homogenize. Then, 70% ethanol was added and the sample including any precipitate was transferred to a RNeasy spin column. This was followed by three washing steps with RW1 and RPE buffers and centrifugation. Finally, RNA was eluted by adding RNase-free water. The following real-time polymerase chain reaction (PCR) was performed at DNA level; therefore, a subsequent conversion of the isolated RNA to cDNA was necessary. For this step, the cDNA synthesis kit from Biozym (Hessisch Oldendorf, Germany) was used. To convert mRNA to cDNA, a master mix containing dNTP Mix, RNase Inhibitor, Oligo(dT)-Primer, cDNA Synthesis Puffer, mRNA and Reverse Transkriptase was prepared. The master mix was then incubated for 60 min at 52.5 °C to continue the reaction with the Reverse Transcriptase. Afterwards, enzymes were inactivated by incubation at 99 °C for 5 min.

### 4.10. Real-Time PCR

Real-time PCR was performed using a Roche Lightcycler nano instrument (Roche, Rotkreuz, Switzerland). The two-step amplification PCR regimen comprised of DNA denaturation at 95 °C, annealing and amplification at 60 °C at a total run of 45 cycles. The reaction was performed in single tubes containing 1 µL cDNA (diluted 1:10 with water), 8 µL water, 10 µL FastStart Essential DNA Probes Master, and 1 mL Primer Assay. Water was added for negative controls instead of cDNA probes. Simultaneous amplification of β-Actin and glyceraldehyde 3-phosphate dehydrogenase (GAPDH) was performed as a control and reference. The primer pairs and the master mix used were purchased from Roche. Finally, the relative expression was determined by using the 2^-ΔΔCt^ method [41]. The Ct-value (cycle threshold) determined at the end of the reaction indicates the cycle number at which the fluorescence signal increases exponentially above a defined threshold for the first time.

### 4.11. Statistical Analysis

For statistical analysis, the SPSS Statistics version 25 (IBM Corp., Armonk, NY, USA) was used. The non-parametric Kruskal–Wallis test was used to compare between and among groups. Correlation analyzes were performed using the Spearman rank correlation coefficient. Kaplan–Meier curves were generated using collected survival data, and differences between these curves were tested by the log-rank test. Evaluation of the cell culture experiments was carried out using the Wilcoxon test. All tests were two-sided, and the level of statistical significance was accepted at *p* ≤ 0.05.

## Figures and Tables

**Figure 1 ijms-21-09287-f001:**
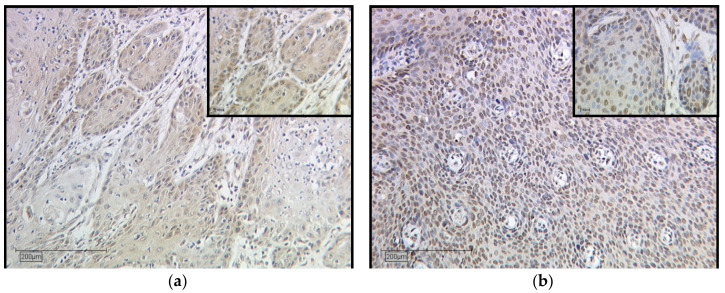
Immunohistochemical staining of LDOC1 in vulvar carcinoma tissue. Representative photomicrographs are presented (10× and 25× magnification): (**a**) cytoplasmatic expression of LDOC1 in vulvar carcinoma; (**b**) nuclear expression of LDOC1 in vulvar carcinoma.

**Figure 2 ijms-21-09287-f002:**
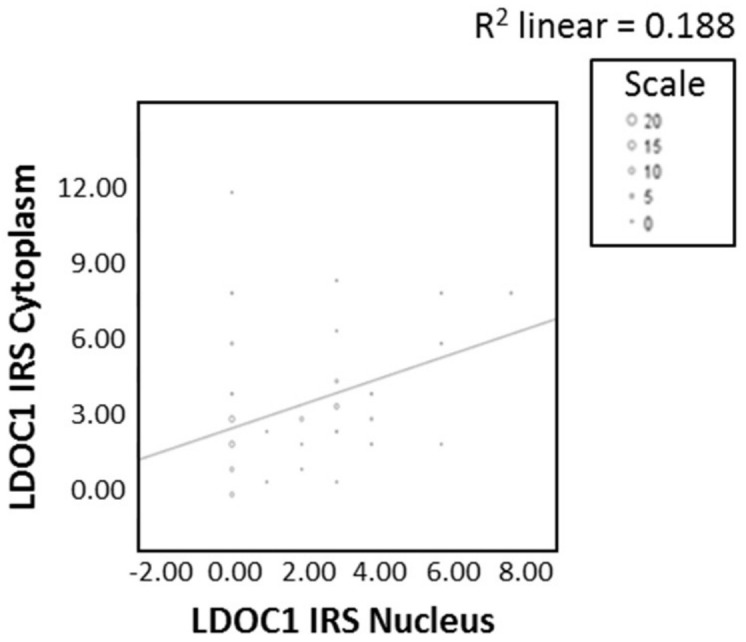
Correlation analysis of cytoplasmatic and nuclear expression of LDOC1. A significant correlation between cytoplasmatic and nuclear expression of LDOC1 was determined by Spearman correlation analysis (ρ = 0.412, *p* = 0.000–0.028).

**Figure 3 ijms-21-09287-f003:**
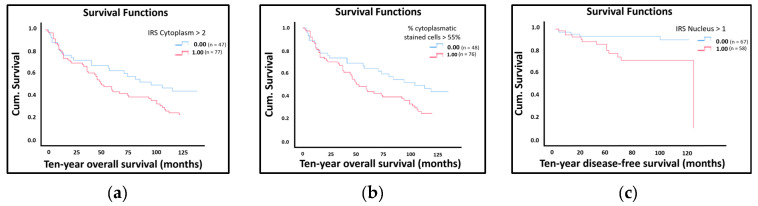
The Kaplan–Meier estimates show that high LDOC1 expression leads to decreased ten-year overall survival/disease-free survival. The red curves represent the event specified in the top-right corner of each graph, while the blue curves represent the opposing event: (**a**) Kaplan–Meier survival univariate analysis for the status cytoplasmatic IRS > 2. Positive expression of LDOC1 in the cytoplasm significantly reduced ten-year overall survival (*p* = 0.049); (**b**) Kaplan–Meier survival univariate analysis for the status cytoplasmatic stained cells >55%. Expression of LDOC1 in 55% of the cytoplasm positively stained cells were significantly related to a worse ten-year overall survival (*p* = 0.046); (**c**) Kaplan–Meier survival univariate analysis for the status nuclear IRS > 1. Positive nuclear staining of LDOC1 showed a significant negative influence on ten-year disease-free survival (*p* = 0.031).

**Figure 4 ijms-21-09287-f004:**
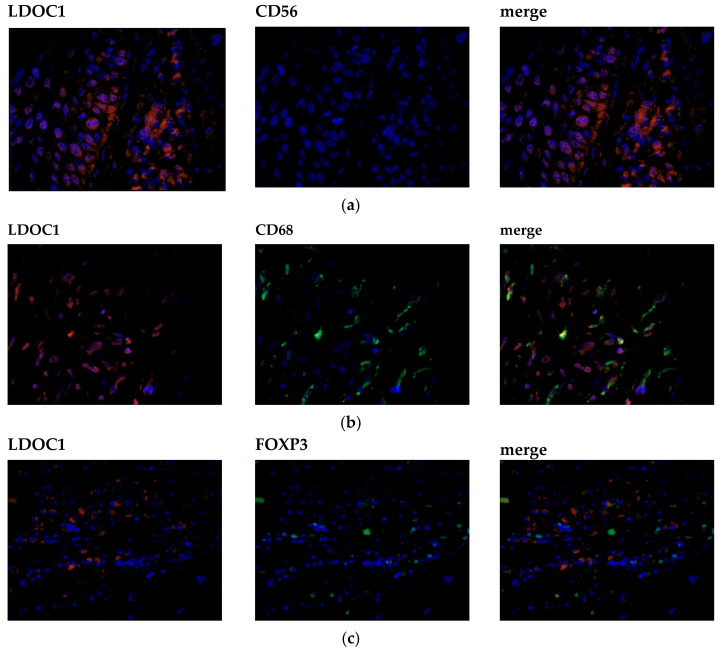
Characterization of the immune cell subpopulation by immunofluorescence double staining. Cell nuclei were marked by DAPI staining (blue). Left column showing LDOC1-positive cells (red), center column showing positive immune cells (green), right column showing a co-expression of the evaluated immune cell marker (green) and LDOC1 (red). Most infiltrating cells were CD68-positive macrophages, followed by FOXP3-positive regulatory T-cells. Representative photomicrographs are presented (40x magnification): (**a**) no expression of CD56-positive immune cells; (**b**) for CD68-positive immune cells (green) a co-expression of LDOC1 (red) could be detected; (**c**) for FOXP3-positive immune cells (green) a co-expression of LDOC1 (red) could be detected.

**Figure 5 ijms-21-09287-f005:**
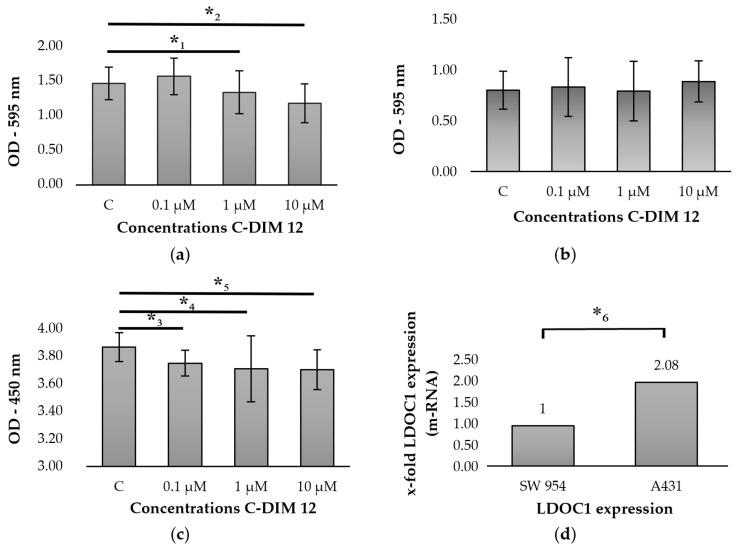
Incubation of cell lines A431 and SW 954 with C-DIM 12: (**a**) MTT assay of cell line A431: cell viability decreases significantly after 72 h of incubation with C-DIM 12 starting at a concentration of 1 μM compared to the unstimulated control cells. Significant observations according to the Wilcoxon test are highlighted with an asterisk (*_1_: *p* = 0.007, *_2_: *p* = 0.005); (**b**) MTT assay of cell line SW 954: no significant influence on cell viability after 72 h of incubation with C-DIM 12 compared to the unstimulated control cells; (**c**) BrdU assay of cell line A431: cell proliferation decreases significantly with increasing C-DIM 12 concentration after 72 h of incubation. Significant observations according to the Wilcoxon test are highlighted with an asterisk (*_3_: *p* = 0.002, *_4_: *p* = 0.003, *_5_: *p* = 0.002); (**d**) real-time PCR analysis of cell lines A431 and SW 954. LDOC1 expression was revealed to be twice as high in A431 in contrast to the cell line SW 954. The difference between the two cell lines was found to be statistically significant according to the Wilcoxon test with *_6_: *p* = 0.026.

**Table 1 ijms-21-09287-t001:** Clinicopathological characteristics of the analyzed vulvar carcinoma samples.

Clinicopathologic Parameters		*n*	Percentage (%)
Histology			
	Keratinizing	160	90.4
	Warty/basaloid	17	9.6
Tumor size			
	T1	69	39
	T2	92	52
	T3	9	5.1
	data unavailable	7	3.9
Nodal status			
	N0	78	44.1
	N1	38	21.5
	N2	12	6.8
	data unavailable	49	27.6
Metastasis			
	M0	8	4.5
	data unavailable	169	95.5
FIGO			
	I	61	34.4
	II	54	30.5
	III	47	26.6
	IV	9	5.1
	data unavailable	6	3.4
Grading			
	G1	29	16.4
	G2	108	61
	G3	39	22
	data unavailable	1	0.6
P16 status			
	Positive	38	21.5
	Negative	57	32.2
	data unavailable	82	46.3

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
