# Peer review of "LDOC1 as Negative Prognostic Marker for Vulvar Cancer Patients"

_ijms, 2020, doi:10.3390/ijms21239287_

Round 1

Reviewer 1 Report

In this manuscript, the authors examined the role of LDOC1 in vulvar cancer. Analysis of patients' sample show that higher cytoplasmic LDOC1 expression is associated with lower overall survival and that nuclear LDOC1 expression is associated with lower survival. However, other results are too preliminary to show the biological role of LDOC1 in vulvar cancer.

  1. Fig. 3. Numbers of each group should be shown.
  2. Fig. 4 shows co-staining of cells with LDOC1 and CD68. Where are cancer cells? Why are cancer cells negative for LDOC1 staining?
  3. Results in Fig. 5 are too preliminary. There is no evidence showing that LDOC1 expression influences sensitivity to NF-kB inhibitor. The effect of C-DIM12 should be compared in SW954 cells with/without LDOC1 overexpression and/or in A431 cells with/without LDOC1 knockdown. The effect of C-DIM12 on LDOC1 expression should also be examined.

Author Response

Revision Details

ijms-984258: LDOC1 as negative prognostic marker for vulvar cancer patients

Reply to Reviewer 1:

First of all, many thanks for your comments on our manuscript. In the following, your comments are in bold print and italic typography, while my answers are not.

In this manuscript, the authors examined the role of LDOC1 in vulvar cancer. Analysis of patients' sample show that higher cytoplasmic LDOC1 expression is associated with lower overall survival and that nuclear LDOC1 expression is associated with lower survival. However, other results are too preliminary to show the biological role of LDOC1 in vulvar cancer.

  1. 3. Numbers of each group should be shown.

The numbers of each group were added to the Figure 3.

  1. 4 shows co-staining of cells with LDOC1 and CD68. Where are cancer cells? Why are cancer cells negative for LDOC1 staining?

In Figure 4 cancer cells and infiltrating immune cells are shown by representative photomicrographs.

We replaced the pictures of Figure 4 b with a better depiction of the tumor cells expressing LDOC1.

In the column on the left, you can see tumor cells and macrophages that express LDOC1 (in red). The picture in the center shows staining of macrophages with CD68 (in green). Comparing this to the latter, it becomes clear that in the first image the macrophages express LDOC1 more intensively than the tumor cells around them. In the merge picture on the right, you can clearly differentiate between the tumor cells (red) and the macrophages (green), that both express LDOC1.

  1. Results in Fig. 5 are too preliminary. There is no evidence showing that LDOC1 expression influences sensitivity to NF-kB inhibitor. The effect of C-DIM12 should be compared in SW954 cells with/without LDOC1 overexpression and/or in A431 cells with/without LDOC1 knockdown. The effect of C-DIM12 on LDOC1 expression should also be examined.

Your assumption is right. Nevertheless, the goal of the test depicted in Figure 5 is to show how the expression of LDOC1 could be regulated by the NF-kB inhibitor C-DIM 12, not the other way around. As a high LDOC1 expression could be identified as negative prognostic marker for vulvar carcinoma patients in the course of this study, we decided to investigate the possibility of a regulation of LDOC1 with the goal of improving the patient’s survival by inhibiting LDOC1. After extensive literature research, we decided to use the NF-κB pathway for it, since it represents a very important transcription factor and is thought to interfere with LDOC1. We therefore didn’t want to silence the LDOC1 expression in vulvar cancer cell lines.

Regarding your comment that the results are too preliminary we agree that further studies focusing on the crosstalk between LDOC1 and the NF-κB pathway are necessary.  But we would suggest a separate study so that its whole complexity can be examined instead of being integrated in a section of this study. The primary goal of this study was to investigate the LDOC1 expression in vulvar carcinoma and its influence on patients’ survival.

Reviewer 2 Report

This is a very nice study on LDOC1 protein in a large set of vulvar carcinoma samples (concerning rarity of this disease).

I recommend accepting the article after minor revision relating to the following points:

  1. Please, comment on the LDOC1 expression in normal vulvar tissue.
  2. I would advise the authors to check for too general sentences which actually do not provide meaningful information, such as:
  • “However, regarding different tumor types studies are controversial.” (line 91) Controversial in what sense?
  • “The literature describes both, the down- as well as the upregulation of LDOC1.” (line 220) The citations are missing just as the information on the site where this down- or up-regulation occurs.
  • “Among other things, a downregulation of LDOC1 has been found in cervical and ovarian cancer.” (line 222). Among what kind of things?
  • “Moreover, sex-specific environmental mechanisms can mediate loss of LDOC1 function as a tumor suppressor. (line 237) What kind of mechanism? This information is unclear.

as well as some minor comments:

  1. regulatory t cells – it should be capitalized - “T” cells.
  2. “In this study we focused on leucine zipper downregulated in cancer 1 (LDOC1), a tumor suppressor candidate gene that was primarily identified to be downregulated in pancreatic and gastric cancer cells [17].” (line 81) the meaning of the word “primarily” here is unclear to me.
  3. “LDOC1 represents a small transcription factor of only 146 amino acids…” (line 83). It is, not “represents”.
  4. Table 1. I guess the word “missing” should be replaced by “not examined” or “data unavailable”.
  5. “The LDOC1 staining process in the cytoplasm resulted in a weak color reaction in 27.6% of the vulvar carcinoma slices (IRS 1-2) (Figure 1a).” (line 106). Replace “slices” with “specimen” or “tissue sections”.
  6. “In order to characterize the immune cells subpopulations, immunofluorescence double staining for LDOC1 and three accepted markers for leucocyte subpopulations was carried out.” (160) I would rather name the markers here. “three accepted markers” sounds a little bit awkward.
  7. “To evaluate a potential regulation of the expression of LDOC1 on vulvar cancer cell lines, MTT- and BrdU-assays were carried out after incubation with C-DIM 12.” (line 183) I think that the name “C-DIM 12” should be explained here, i.e. where it appears first in the text.
  8. “Since LDOC1 is ubiquitously expressed in normal tissue but is downregulated in many tumors, the hypothesis of LDOC1 as a tumor suppressor gene has been formulated by Nagasaki et al. [17].” (line 220) Please, explain which normal tissues express LDOC1.
  9. “However, in the patient population underlying this study, less than half of the tissue samples expressed LDOC1 in the nucleus.” (line 225) The word “underlying” sounds awkward here.
  10. “Duzkale et al. revealed that upregulation of LDOC1 in unmutated CLL was associated with a poor prognosis [18].” (line 228) The abbreviation CLL should be explained here.

Author Response

Revision Details

ijms-984258: LDOC1 as negative prognostic marker for vulvar cancer patients

Reply to Reviewer 2:

First of all, many thanks for your comments on our manuscript. In the following, your comments are in bold print and italic typography, while my answers are not.

This is a very nice study on LDOC1 protein in a large set of vulvar carcinoma samples (concerning rarity of this disease).

I recommend accepting the article after minor revision relating to the following points:

  1. Please, comment on the LDOC1 expression in normal vulvar tissue.

We added representative pictures of immunohistochemical staining of normal vulvar tissue and vulvar intraepithelial neoplasia (VIN) in the supplementary file. In normal vulvar tissue no LDOC1 expression was shown, whereas moderate cytoplasmatic LDOC1 expression could be detected in vulvar intraepithelial neoplasia.

  1. I would advise the authors to check for too general sentences which actually do not provide meaningful information, such as:
  • “However, regarding different tumor types studies are controversial.” (line 91) Controversial in what sense?

The main message of this section was to show that LDOC1 had been described as positive as well as negative prognostic marker in the literature depending on the examined tumor type. Having mentioned studies in which LDOC1 was described as downregulated in the first text section, we wanted to point out that other studies examining other tumor types showed an opposite result. We added examples and citations of the named studies.

You will find the adjustment in line 91-93:

Studies examining other tumor identities such as chronic lymphatic leukemia or head and neck squamous cell carcinoma come to opposite conclusions. Those studies have reported that LDOC1 overexpression increases cell proliferation and is correlated with a poor prognosis.

  • “The literature describes both, the down- as well as the upregulation of LDOC1.” (line 220) The citations are missing just as the information on the site where this down- or up-regulation occurs.

Your observation is right. We decided to remove this sentence as the section starting in line 91 treats this topic sufficiently. It was originally intended to point out that LDOC1 had been described as both a positive and negative prognostic marker.

You will find the adjustment in line 225.

  • “Among other things, a downregulation of LDOC1 has been found in cervical and ovarian cancer.” (line 222). Among what kind of things?

The other tumor identities besides cervical and ovarian cancer are now listed with citations.

You will find the adjustment in line 227-228:

In addition to tumor identities like pancreatic, esophageal, and colorectal cancer, a downregulation of LDOC1 has also been found in cervical and ovarian cancer.

  • “Moreover, sex-specific environmental mechanisms can mediate loss of LDOC1 function as a tumor suppressor. (line 237) What kind of mechanism? This information is unclear.

We refer to Duzkale et al. who revealed in a recent study from 2019, that by comparing saliva samples from patients with oral squamous cell carcinoma (OSCC), a gender-specific difference in LDOC1 expression was noticed: while LDOC1 expression levels were low in males, LDOC1 expression levels were considerably higher in female patients. They concluded that salivary LDOC1 is a gender-difference biomarker of OSCC. Hence, the authors formulated the hypothesis that this observation could have something to do with gender-different smoking habits. The authors refer to future work, in which an in vivo investigation should examine how the difference in LDOC1 expression levels between the genders contributes to the various mechanisms underlying tumorigenesis.

You will find the adjustment in line 244.

as well as some minor comments:

  1. regulatory t cells – it should be capitalized - “T” cells.

Done.

  1. “In this study we focused on leucine zipper downregulated in cancer 1 (LDOC1), a tumor suppressor candidate gene that was primarily identified to be downregulated in pancreatic and gastric cancer cells [17].” (line 81) the meaning of the word “primarily” here is unclear to me.

You will find the adjustment in line 82-83:

In this study we focused on leucine zipper downregulated in cancer 1 (LDOC1), a tumor suppressor candidate gene that was discovered for the first time to be downregulated in pancreatic and gastric cancer cells.

  1. “LDOC1 represents a small transcription factor of only 146 amino acids…” (line 83). It is, not “represents”.

Done.

  1. Table 1. I guess the word “missing” should be replaced by “not examined” or “data unavailable”.

Done.

  1. “The LDOC1 staining process in the cytoplasm resulted in a weak color reaction in 27.6% of the vulvar carcinoma slices (IRS 1-2) (Figure 1a).” (line 106). Replace “slices” with “specimen” or “tissue sections”.

Done.

  1. “In order to characterize the immune cells subpopulations, immunofluorescence double staining for LDOC1 and three accepted markers for leucocyte subpopulations was carried out.” (160) I would rather name the markers here. “three accepted markers” sounds a little bit awkward.

You will find the adjustment in line 165-166:

In order to characterize the immune cells subpopulations, immunofluorescence double staining for LDOC1 and three markers for leucocyte subpopulations CD56, CD68 and FOXP3 was carried out.

  1. “To evaluate a potential regulation of the expression of LDOC1 on vulvar cancer cell lines, MTT- and BrdU-assays were carried out after incubation with C-DIM 12.” (line 183) I think that the name “C-DIM 12” should be explained here, i.e. where it appears first in the text.

You will find the adjustment in line 189-190:

To evaluate a potential regulation of the expression of LDOC1 on vulvar cancer cell lines, MTT- and BrdU-assays were carried out after incubation with the NF-кB inhibitor C-DIM 12 (3,3'-[(4-Chlorophenyl)methylene]bis[1H-indole]).

  1. “Since LDOC1 is ubiquitously expressed in normal tissue but is downregulated in many tumors, the hypothesis of LDOC1 as a tumor suppressor gene has been formulated by Nagasaki et al. [17].” (line 220) Please, explain which normal tissues express LDOC1.

You will find the adjustment in line 225-226:

Since LDOC1 is ubiquitously expressed in normal tissue as brain, thyroid, heart, kidney and pancreas tissue but is downregulated in many tumors, the hypothesis of LDOC1 as a tumor suppressor gene has been formulated by Nagasaki et al.

  1. “However, in the patient population underlying this study, less than half of the tissue samples expressed LDOC1 in the nucleus.” (line 225) The word “underlying” sounds awkward here.

You will find the adjustment in line 231:

However, in the patient population of this study, less than half of the tissue samples expressed LDOC1 in the nucleus.…

  1. “Duzkale et al. revealed that upregulation of LDOC1 in unmutated CLL was associated with a poor prognosis [18].” (line 228) The abbreviation CLL should be explained here.

Done.

Round 2

Reviewer 1 Report

Additional experiments were not been performed. The roles of LDOC1 in immune cells and cancer cells and the relationship between LDOC1 and NF-kB signaling are still unclear.  

Author Response

First of all, many thanks for your comments on our manuscript. In the following, your comments are highlighted by bold italics.

Additional experiments were not been performed. The roles of LDOC1 in immune cells and cancer cells and the relationship between LDOC1 and NF-kB signaling are still unclear.  

The primary goal of this study was to investigate the LDOC1 expression in vulvar carcinoma and its influence on patients’ survival, since no information about the expression and involvement of LDOC1 in vulvar cancer exists to date. Furthermore, previous studies with LDOC1 regarding different tumor types are controversial whether it is a positive or negative prognostic marker. In the course of this study, a high LDOC1 expression in vulvar cancer cells could be identified as negative prognostic marker for vulvar carcinoma patients reducing patients’ overall and disease-free survival.

Assessing the vulvar carcinoma samples, we found that in almost all samples the immune cells were expressing LDOC1. Unfortunately, neither the correlation analyses nor the Kaplan-Meier curves did show any influence on the patients’ survival. Infiltrating immune cells can either inhibit or promote tumor progression depending on their type. Consequently, we wanted to characterize the immune cells that expressed LDOC1. The investigation by immunofluorescence double staining showed that CD68 positive macrophages were the main part of the LDOC1 positive immune cell infiltrate, followed by FOXP3 positive regulatory T cells. Since macrophages can have a negative influence on the tumor, it is plausible that they express LDOC1, as it was identified as a negative prognostic marker for vulvar cancer in this study. Since very little is known about LDOC1 and its expression in immune cells, our study serves the purpose of presenting a general assessment to this hypothesis. Whether expression of LDOC1 in the tumor attracts mainly macrophages, or whether the tumor suppressor LDOC1 is expressed independently in the immune cells, must be investigated in further experiments.

As a high LDOC1 expression could be identified as negative prognostic marker for vulvar carcinoma patients in the course of this study, we decided to investigate the possibility of a regulation of LDOC1 with the goal of improving the patients’ survival by inhibiting LDOC1. After extensive literature research, we decided to use the NF-κB pathway for it, since it represents a very important transcription factor that is activated and associated with poor prognosis in many tumor identities. As also cited in the manuscript activation of NF-κB could also be detected in the nucleus of vulvar cancer cells. Furthermore, a study by Song et al. revealed that upon LDOC1 transfection of human intrahepatic biliary epithelial cells (HIBECs), the mRNA rate of NF-κB increased significantly while the apoptosis rate decreased. Since the signaling pathway around NF-κB plays a crucial role in vulvar carcinomas and is regulated by LDOC1, the effect of the NF-κB inhibitor C-DIM 12 was investigated in the present study on two vulvar carcinoma cell lines. We agree that further studies focusing on the crosstalk between LDOC1 and the NF-κB pathway are necessary.  We would suggest a separate study in order to examine in detail the entire complexity instead of just integrating it in a section of this study.